# Prediction of Adverse Post-Infarction Left Ventricular Remodeling Using a Multivariate Regression Model

**DOI:** 10.3390/diagnostics12030770

**Published:** 2022-03-21

**Authors:** Valentin Oleynikov, Lyudmila Salyamova, Olga Kvasova, Nadezhda Burko

**Affiliations:** Department of Therapy, Medical Institute, Penza State University, 440026 Penza, Russia; v.oleynikof@gmail.com (V.O.); l.salyamova@yandex.ru (L.S.); olgkvasova@rambler.ru (O.K.)

**Keywords:** myocardial infarction, adverse left ventricular remodeling, echocardiography, left ventricular-arterial coupling

## Abstract

Background. In order to provide personalized medicine and improve cardiovascular outcomes, a method for predicting adverse left ventricular remodeling (ALVR) after ST-segment elevation myocardial infarction (STEMI) is needed. Methods. A total of 125 STEMI patients, mean age 51.2 (95% CI 49.6; 52.7) years were prospectively enrolled. The clinical, laboratory, and instrumental examinations were performed between the 7th and 9th day, and after 24 and 48 weeks, including plasma analysis of brain natriuretic peptide (BNP), transthoracic echocardiography, analysis of left ventricular-arterial coupling, applanation tonometry, ultrasound examination of the common carotid arteries with RF signal amplification. Results. Patients were divided into 2 groups according to echocardiography: “ALVR” (*n* = 63)—end-diastolic volume index (EDVI) >20% and/or end-systolic volume index (ESVI) >15% after 24 weeks compared with initial values; “non-ALVR” (n = 62)—EDVI <20% and ESVI <15%. In the ALVR group, hard endpoints (recurrent myocardial infarction, unstable angina, hospitalization for decompensated heart failure, ventricular arrhythmias, cardiac surgery, cardiovascular death) were detected in 19 people (30%). In the non-ALVR group, hard endpoints were noted in 3 patients (5%). The odds ratio of developing an adverse outcome in ALVR vs. non-ALVR group was 8.5 (95% CI 2.4–30.5) (*p* = 0.0004). According to the multivariate analysis, the contribution of each of the indicators to the relative risk (RR) of adverse cardiac remodeling: waist circumference, RR = 1.02 (95% CI 1.001–1.05) (*p* = 0.042), plasma BNP—RR = 1.81 (95% CI 1.05–3.13) (*p* = 0.033), arterial elastance to left ventricular end-systolic elastance (Ea/Ees)—RR = 1.96 (95% CI 1.11–3.46) (*p* = 0.020). Conclusion. Determining ALVR status in early stages of the disease can accurately predict and stratify the risk of adverse outcomes in STEMI patients.

## 1. Introduction

Chronic heart failure (CHF) is one of the main causes of hospitalization and mortality in the world, including the Russian Federation [1,2]. The progressive nature of the course, early disability, the need for long-term pharmacotherapy and cardiac surgery are associated with significant economic costs for the healthcare system.

The results of a meta-analysis conducted by N.R. Jones et al. showed a significant improvement in the survival rates of patients with CHF from the 1970s to the 1990s. However, mortality has declined slightly over the past two decades. In 2010–2019, one-year and five- year survival amounted to 89.3% (84.3–93.4%) and 59.7% (54.7–64.6%), respectively [3]. Another study showed five times increase in the risk of a fatal outcome in the development of CHF [4].

According to the American Heart Association report, CHF prevalence is predicted to increase by 46% in the USA by 2030 [5]. CHF often complicates the postinfarction period. According to the results of the Russian EPOCHA-CHF study, along with arterial hypertension and chronic ischemic heart disease, myocardial infarction (MI) has become a competing cause of CHF, accounting for 15.8% in 2017 vs. 5.8% in 1998 [6]. The results of the European Society of Cardiology Heart Failure Long-Term Registry (ESC-HF-LT-R) also demonstrate the prevailing role of coronary heart disease in the development of CHF [7].

The widespread introduction of high-tech methods for the treatment of acute MI has led to a decrease in hospital mortality; this has been accompanied, however, by an increase in the number of patients with CHF [1,6]. Due to the very typical prolongation of myocardial revas cularization terms, adverse postinfarction remodeling often develops being a key pathogenetic element of CHF and a significant predictor of mortality.

Adverse postinfarction left ventricular remodeling (ALVR) is characterized by an increase in the end-diastolic volume (EDV) >20% or the end-systolic volume (ESV) >15% compared with baseline values [8]. Ventricular remodeling already occurs within the first hours after cardiomyocyte necrosis and proceeds for several months. This process is characterized by a change in the left ventricular (LV) shape and size, and its dysfunction [8,9]. Early remodeling develops within three months after acute MI; mid-term and late remodeling develop within six and twelve months, respectively [10].

Identification of patients with a high probability of ALVR development in the early stages of the disease is important for the stratification of cardiovascular risk, the choice of personalized anti-remodeling therapy and rehabilitation.

The aim of this study was to search for early predictors and develop a model for predicting adverse remodeling in patients who had ST-segment elevation myocardial infarction (STEMI).

## 2. Materials and Methods

An open prospective single-center study involving 141 STEMI patients was con- ducted at the Department of Therapy of Penza State University (Penza, Russia). The study protocol and informed consent were approved by the Local Ethics Committee at Penza State University (approval code 317, on 15 May 2020).

The study included patients which met the following criteria: aged 35–65 years; acute STEMI, confirmed by an electrocardiogram, a diagnostically significant increase in specific cardiac enzymes (troponin I, CPK-MB); the presence of hemodynamically significant stenosis of the infarct-related artery according to coronary angiography with occlusion of other coronary arteries less than 50%, including the trunk of the left coronary artery less than 30%. The main exclusion criteria included: repeated or recurrent MI; type 1 or type 2 diabetes (requiring insulin therapy); NYHA class II-IV of CHF, severe concomitant diseases.

The treatment of STEMI patients has been carried out in full accordance with the guidelines [11] over the entire follow-up period.

A comprehensive clinical, laboratory and instrumental examination has been conducted with preserved pharmacotherapy initially in the period from the 7th to the 9th day of the STEMI, and after 24- and 48-week follow up (Figure 1).

Plasma brain natriuretic peptide (BNP) (with EDTA-ethylenediaminetetraacetic acid) has been measured using the OLYMPUS AU400 chemistry analyzer (Olympus Corporation, Tokyo, Japan).

Transthoracic echocardiography (EchoCG) has been performed using the MyLab90 ultrasound scanner (Esaote, Genoa, Italy) for determination of standard parameters and subsequent calculation of indexed values for the end-diastolic volume (EDVI) and the end- systolic volume (ESVI). An increase in EDVI >20% and/or ESVI >15% has been taken as ALVR after 24 weeks compared with the initial values (7–9th day). The interaction of the left ventricle and the arterial bed has been analyzed according to the following parameters: LV end-systolic elastance, which is the ratio of the end-systolic pressure to the end-systolic volume and reduced to body surface area (Ees/BSA); the arterial elastance calculated as the ratio of the end-systolic pressure over stroke volume divided by body surface area (Ea/BSA); LV-arterial coupling (LVAC) index, defined as the ratio of arterial elastance to left ventricular end-systolic elastance (Ea/Ees) [12].

The structural and functional state of the common carotid arteries (CCA) has been assessed with the MyLab ultrasound machine (Esaote, Genoa, Italy) using radiofrequency (RF)- based technology. The following indicators were recorded: quality intima-media thickness (QIMT); stiffness index β; local systolic arterial pressure in CCA (loc Psys); local diastolic arterial pressure in CCA (loc Pdia). The above indicators of local pressure and stiffness are calculated using special software based on the level of blood pressure in the brachial artery, changes in the diameter and volume of the CCA in systole and diastole [13].

Applanation tonometry has been used to determine systolic aortic pressure (SBPao) and diastolic (DBPao) aortic pressure, and carotid-femoral pulse wave velocity (cfPWV) using the SphygmoCor device (AtCor Medical, Sydney, Australia) [14].

### Statistical Analysis

Statistical data processing was performed using the licensed version of STATISTICA 13.0 program (StatSoft, Inc., Tulsa, OK, USA). All indicator values were given with the 95% confidence interval (CI). The dynamics of indicators was analyzed by the method of one-way analysis of variance (ANOVA) using the Newman-Keuls test. The Cox multiple linear regression was used when constructing a multivariate model. The frequency of the endpoint development was determined by the Kaplan-Meier method. The level of statistical significance was *p* < 0.05.

## 3. Results

We have summed up a 48-week follow-up for 125 patients (88.7%). There were some reasons for early termination of study participation: 1 patient died due to myocardial rupture on the 16th day; 1 patient died from pulmonary edema (according to the autopsy report) at the 10th month; 3 patients moved to another city; 11 patients discon tinued the follow-up due to low adherence.

The age of the patients included in the study was 51.2 (49.6; 52.7) years, men prevailed—109 patients (87.2%). The body mass index was 27.4 (26.7; 28.0) kg/m^2^. Symptoms of abdominal obesity were diagnosed in 74 patients (59.2%) [14]. Coronary heart disease was noted in 21 patients in the history (16.8%). Heredity cardiovascular diseases burdened 51 patients (40.8%) and 80 patients were smokers (64%). Arterial hypertension was observed in 61.6% (n = 77) with a disease duration of 6.5 (5.2; 7.8) years. The mean level of systolic blood pressure (SBP) was 118.8 (116.3; 121.3) mmHg and that of diastolic blood pressure (DBP)—76.2 (74.6; 77.9) mmHg. Prior to the STEMI, antihypertensive therapy was received regularly by 16 patients (20.8%), and irregularly by 24 patients (31.2%); 37 patients (48%) were not treated.

Primary percutaneous coronary intervention has been performed in 56 patients (44.8%); pharmacoinvasive strategy has been performed in 68 cases (54.4%). One patient only received thrombolytic therapy at the pre-hospital stage.

The analysis of EDVI and ESVI dynamics within 24 weeks after acute myocardial infarction has made it possible to divide the patients into 2 groups. The first group included 63 patients with ALVR revealed after 24 weeks according to the echocardiography. The second group included 62 patients without ALVR (non-ALVR group). A comparative analysis of patients by age, some anthropometric and anamnestic indicators and therapy is presented in Table 1.

The study of echocardiographic parameters in patients with ALVR revealed a progressive increase in EDVI and ESVI during the entire follow-up period. Thus, within 24–48 weeks, EDVI increased by 22.3–21.1%, and ESVI by 26.9–25.7%, respectively (Figure 2, Appendix A). Besides, negative dynamics of ejection fraction (EF) has been noted during the repeated studies. At the same time, some negative dynamics of ESVI were accompanied by an increase in EF by 3.4% in patients without ALVR by the end of the follow-up. Differences in the presented echocardiographic parameters of the compared groups were noted for 48 weeks (Figure 1).

Patients with adverse LV remodeling and without it initially varied by BNP level: 231.9 (95% CI 122.9; 340.9) vs. 72.1 (95% CI 51.0; 93.2) (*p* = 0.003). Despite the improvement in laboratory values for each group, the differences persisted at subsequent visits (Figure 3).

In the analyzed groups, the inverse dynamics of LVAC indicators were revealed. Initially, the patients of two groups had comparable values of arterial elastance. Ea/BSA indicator in ALVR group decreased after 24 weeks and returned to the baseline after 48 weeks; in non-ALVR group, it decreased by the end of the follow-up (Table 2).

Initially, in patients with ALVR, the level of LV elastance was significantly reduced, and LVAC index predominated compared to non-ALVR group. After 24 and 48 weeks, a decrease in Ees/BSA by 18.6–14%, and an increase in Ea/Ees were denoted by the end of the follow-up in ALVR group. At the same time in the comparison group the LV elastance has not changed, and LVAC index decreased by 10.6% after 24 weeks and by 16% after 48 weeks. Intergroup differences in Ees/BSA and Ea/Ees values persisted over the entire follow-up period.

A detailed analysis of LVAC index has elicited abnormal values in ALVR group: initially in 29 patients (46%), then in 36 patients (57%; unreliable) after 24 weeks, and finally in 33 patients (52.4%; unreliable) after 48 weeks. As for non-ALVR group, there were initially 7 patients (11.3%), and 2 patients (3%; *p* = 0.08) at the interval visit with abnormal LVAC values. By the end of the follow-up, all of the patients of the comparison group demonstrated a normal level of LVAC index (0%; *p* = 0.008).

An analysis of applanation tonometry indicators has shown low SBPao values in ALVR group. Within 48 weeks, a comparable increase in central pressure parameters in the comparison groups has been found. In patients with ALVR and without it cfPWV has not shown differences and was stable over the entire follow-up period.

According to the ultrasound of CCA using the RF-technology, the groups had a com- parable QIMT level and β index in the period from the 7th to the 9th day of the STEMI. In ALVR group, indicators have decreased after 24 and 48 weeks; in non-ALVR group, QIMT values have also improved during therapy.

A comparative analysis of the dynamics of local pressure in CCA is essential. The level of loc Psys and loc Pdia has been initially reduced in patients with ALVR compared with non-ALVR group. Subsequently, the pressure values in CCA in ALVR group were restored after 24–48 weeks.

One of the important results of this study is the relationship between ALVR and cardiovascular events. In the comparison groups, the frequency of hard endpoints, such as recurrent MI, unstable angina, hospitalization for decompensated heart failure, ventricu lar arrhythmias, cardiac surgery, and death from cardiac causes has been analyzed (Figure 4). In ALVR group, the above endpoints were detected in 19 patients (30%): 7 patients were hospitalized for unstable angina (11.1%); 1 patient (1.6%) was diagnosed with recur- rent acute MI and 2 patients (3.2%) due to decompensated CHF. Cardiac surgery was performed in 7 patients (11.1%). Moreover, 2 patients were diagnosed with life-threatening arrhythmias (3.2%). In non-ALVR group, hard endpoints were noted in 3 patients (5%); 1 patient underwent cardiac surgery (1.6%), 2 patients (3.2%) were hospitalized for unstable angina. The odds ratio of developing an adverse outcome in group 1 compared with group 2 was 8.5 [95% CI 2.4–30.5] (*p* = 0.0004).

The predictors of adverse postinfarction remodeling were identified according to the univariate and multivariate logistic regression analysis of clinical, laboratory and instrumental parameters recorded in the period of the 7th-9th days from the index event. Since the determination of postinfarction remodeling is based on echocardiographic parameters, the latter were excluded from the analysis [8]. According to the univariate analysis, the independent variables of ALVR were waist circumference (WC), determination of BNP in a bimodal distribution (“0”—with BNP <100 pg/mL; “1”—BNP ≥100 pg/mL), EF, Ees/BSA, Ea/Ees, Ea/Ees in bimodal distribution (“0”—at Ea/Ees 0.6–1.2; “1”—at Ea/Ees <0.6 or >1.2), Loc Psys (Table 3).

A multivariate model for the development of various types of postinfarction remodeling including WC, abnormal values of BNP and Ea/Ees was created based on the results of the univariate analysis and considering the correlations between the indicators (Table 4).

The multivariate regression model presented in Table 4 uses the formula (1):


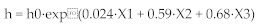
(1)
where: X1 is WC, cm; X2 is equal to 1.0 with BNP ≥100 pg/mL, and it is equal to 0 with BNP <100 pg/mL; X3 is equal to 1.0 with Ea/Ees ranging between 0.6 and 1.2, and it is equal to 0 with a normal value of Ea/Ees; h0(t) is baseline risk of 0.024329 at the 24th week after STEMI. If the value of h is higher than 1.0, the development of ALVR is predicted; if the value of h is less than 1.0, a conclusion is made about the absence of ALVR.

The created model for determining the risk of developing the postinfarction remodeling has shown good informative results: the Wilks’ lambda = 0.65256; F (3.105) = 18.63467 (*p* < 0.00001).

## 4. Discussion

The development of CHF post-MI is driven by the complex pathophysiological mechanisms underlying cardiac remodeling: an inflammatory reaction in the area of myocardial necrosis, isolation of intracellular signaling proteins, activation of neurohumoral systems, followed by the development of hypertrophy and cardiac dilatation, and the formation of a connective tissue scar.

The structural and functional remodeling of LV was followed by a decrease in its contractile function leading to impairment of hemodynamics in organs and tissues [15,16]. The adverse cardiac remodeling post-MI leads to CHF development, associated with increased re-hospitalization rate, disability, and mortality of patients [10,17]. The present study demonstrates that adverse postinfarction remodeling is associated with a high risk of cardiovascular events, being 8.5 times higher than that compared with non-ALVR group during the 48-week follow-up. Thus, the authors have developed a model for predicting various forms of cardiac remodeling to timely prescribe medication and conduct dynamic monitoring for patients with a high risk of an adverse outcome.

Traditional factors have little effect on prognosis after STEMI [18]. In our study, the patients who subsequently developed different types of postinfarction remodeling did not experience differences in most risk factors, except for WC. In this connection, a search for new predictors of unfavorable structural and functional cardiac remodeling seems to be highly relevant.

The high frequency of thrombolytic therapy as part of the pharma-coinvasive treatment strategy should be noted; this was present in 54.4% of cases, which is due to the late presentation of the patients and the territorial remoteness of the place of residence of the patients from the hospital.

In accordance with the clinical practice guidelines for CHF, the determination of natriuretic peptides is used in diagnosing this complication and in assessing the prognosis [2,19]. A high level of N-terminal pro-brain natriuretic peptide (NT-proBNP) indicates an increasing risk of sudden death, recurrent MI, and CHF in patients with MI and unstable angina [20]. The level of BNP was significantly higher in patients with ALVR as compared to non-ALVR group over the entire follow-up period. As a result, the indicator was included in both univariate and multivariate models for predicting adverse postinfarction cardiac remodeling.

The development and progression of cardiovascular diseases are associated with deterioration in the structural and functional properties of the vascular wall, being an important predictor of an adverse outcome, regardless of traditional factors [21]. According to Lechner I. et al., an enhanced level of pulse wave velocity (PWV) in STEMI patients predicted the development of cardiac and cerebrovascular adverse events 13 months after the index event. In another study, cfPWV appeared to be an important predictor of recovery of LV contractile function three and six months after STEMI [22].

An increased aortic stiffness causes impairment of the coronary blood flow and the development of ischemia of the subendocardial layer even without the coronary artery stenosis. Early return of the reflected pulse wave is accompanied by an increase in SBP and decrease in DBP. The LV load and myocardial oxygen demand increase, but perfusion pressure deteriorates, and myocardial ischemia develops due to lower DBP [23]. Besides, a decrease in the damping function of the aorta, combined with an increase in total peripheral vascular resistance, significantly reduces the efficiency of LV contraction [24].

In this study, the compared groups initially differed in the level of SBPao, and SBP and DBP in CCA. Loc Psys has only shown predictive value in the development of ALVR in univariate logistic regression analysis. However, the indicators of the structural and functional state of the aorta and CCA have not been included in the multivariate model.

The functioning of the cardiovascular system as a whole is determined by the adequacy of the interaction between the heart and the arterial system during the ejection of blood from the LV and is called LVAC [25]. The predictive significance of this parameter has been demonstrated in a number of studies. In particular, in patients with ischemic cardiomyopathy, the ratio Ea/Ees < 1.47 was characterized by better survival rate compared to those whose indicator exceeded the specified threshold value [12]. LVAC can be used both to clarify cardiovascular risk and to study the efficiency of treatment.

LVAC indicator is calculated as the ratio of arterial elastance (Ea) to LV end-systolic elastance (Ees). The Ea parameter indicates the arterial load exerted on LV during the blood ejection, regardless of its functional ability. Arterial afterload includes aortic valve resistance, systemic vascular resistance (SVR), arterial capacitance and stiffness, and duration of systole and diastole. The Ees parameter indicates LV contractility and systolic stiffness. In our study, a low level of Ees/BSA and higher LVAC values were initially diagnosed in the ALVR group. These differences have remained over the entire follow-up period. Besides, these indicators of LV-arterial interaction have evidenced predictive significance according to univariate regression analysis. The determination of the normal/abnormal LVAC level was adequate for inclusion in the mul tivariate model based on the studied parameters.

With the results obtained, a complex model for predicting ALVR in the period from the 7th to the 9th day of the STEMI was developed based on the analysis of WC, BNP, and LVAC.

## 5. Conclusions

In the present study, the development of adverse postinfarction LV remodeling is associated with an 8.5-fold increase in the risk of cardiovascular events. The following independent factors of adverse cardiac remodeling were determined 24 weeks after STEMI: waist circumference; abnormal values of brain natriuretic peptide; end-systolic left ventricular elastance normalized to body sur- face area; abnormal level of Ea/Ees index; local systolic pres sure in the common carotid arteries.

Based on the results of multivariate regression analysis, a model for predicting various types of postinfarction remodeling based on waist circumference, abnormal values of brain natriuretic peptide, and the LV arterial coupling index has been developed.

## 6. Study limitations

The study was conducted in patients with single-vessel lesion of the coronary bed according to coronary angiography: the presence of hemodynamically significant stenosis of only the infarct-related artery with occlusion of other coronary arteries less than 50%, including the trunk of the left coronary artery—less than 30%. Moreover, the proposed multivariate model was developed for patients with primary STEMI aged 35 to 65 years and is not applicable to patients with recurrent MI, as well as MI without ST segment elevation, younger or older than this age.

## Figures and Tables

**Figure 1 diagnostics-12-00770-f001:**
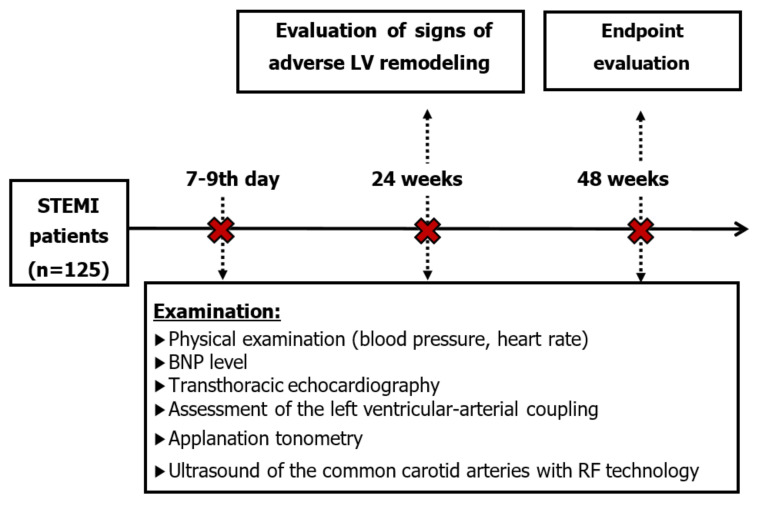
Flowchart of the study.

**Figure 2 diagnostics-12-00770-f002:**
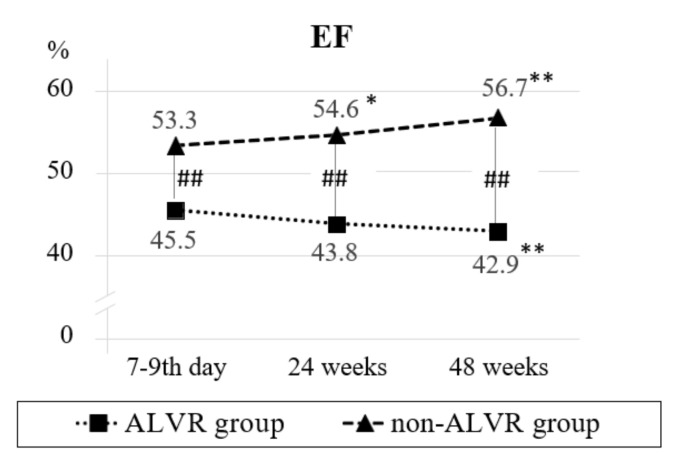
Dynamics of EF values in the comparison groups. Note: * *p* < 0.05, ** *p* < 0.01 are significant differences between the initial values and subsequent visits; ## *p* < 0.01 are significant intergroup differences. EF—ejection fraction.

**Figure 3 diagnostics-12-00770-f003:**
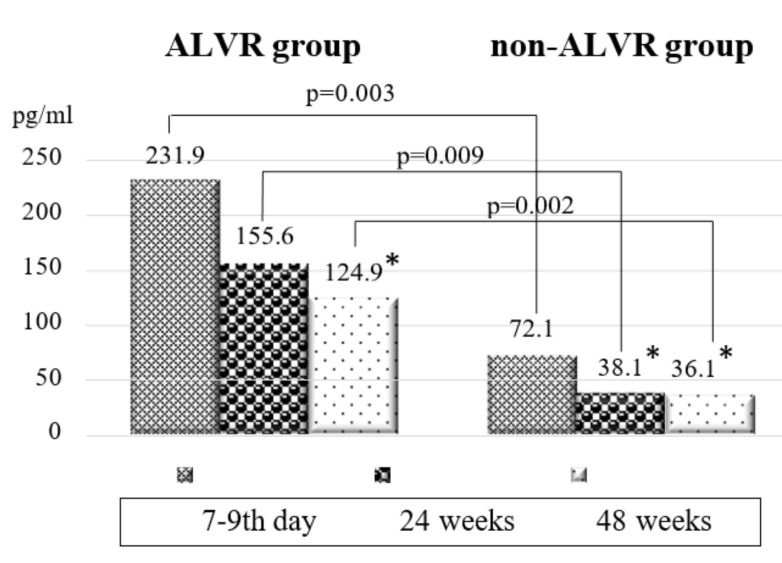
Dynamics of BNP in the comparison groups. Note: * *p* < 0.05 are significant differences between the initial values and subsequent visits. BNP –brain natriuretic peptide.

**Figure 4 diagnostics-12-00770-f004:**
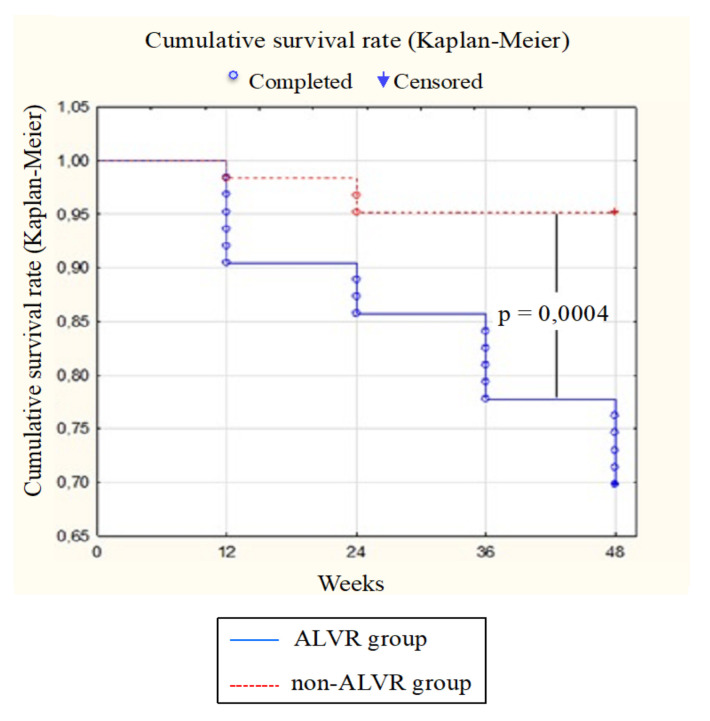
The incidence of hard endpoints in the comparison groups.

**Table 1 diagnostics-12-00770-t001:** A comparative analysis of ALVR and non-ALVR groups (n = 125).

Indicators	ALVR Group (n = 63)	non-ALVR Group (n = 62)	*p*
Age, years	51.4 (49.2; 53.6)	50.9 (48.7; 53.1)	0.724
Female, n (%)	9 (14.3%)	6 (9.7%)	0.246
Male, n (%)	54 (85.7%)	56 (90.3%)	0.246
Abdominal obesity, n (%)	41 (65%)	33 (53.2%)	0.086
Waist circumference (WC), cm	99.1 (96.4; 101.9)	92.9 (90.1; 95.6)	**0.002**
BMI, kg/m^2^	28 (27.1; 28.9)	26.7 (25.8; 27.6)	0.056
Tobacco smoking, n (%)	38 (60.3%)	42 (67.7%)	0.176
Smoking history, years	26.4 (23.4; 29.4)	27.4 (24.6; 30.3)	0.619
Burdened heredity, n (%)	27 (42.8%)	24 (38.7%)	0.325
History of CHD, n (%)	11 (17.5%)	10 (16.1%)	0.383
CHD duration, years	2.4 (0; 4.9)	2.8 (0.2; 5.5)	0.798
AH, n (%)	37 (58.7%)	40 (64.5%)	0.245
AH duration, years	7.6 (5.8; 9.5)	5.4 (3.7; 7.2)	0.090
SBP, mmHg	118.1 (114.5; 121.6)	119.4 (115.9; 122.9)	0.586
DBP, mmHg	76.6 (74.2; 78.9)	75.9 (73.6; 78.2)	0.696
HR, bpm	71.1 (69.4; 72.8)	69.9 (68.2; 71.5)	0.305
	Drug therapy		
Dual antiplatelet therapy, n (%)	63 (100%)	62 (100%)	0.500
Statins, n (%)	63 (100%)	62 (100%)	0.500
Beta blockers, n (%)	56 (89%)	51 (82%)	0.133
ACE (angiotensin converting enzyme) inhibitors/sartans, n (%)	49 (78%)	53 (86%)	0.122
Calcium channel block-ers, n (%)	5 (8%)	5 (8%)	0.500
Diuretics, n (%)	12 (19%)	10 (16%)	0.329

Note: the data are presented as M ± SD with a normal distribution, and as Me (Q25%; Q75%) with an incorrect distribution; n is the number of patients; BMI is body mass index; CHD is coronary heart disease; AH is arterial hypertension; SBP is systolic blood pressure; DBP is diastolic blood pressure; HR is heart rate.

**Table 2 diagnostics-12-00770-t002:** Comparative characteristics of LVAC indicators, and structural and functional features of arteries in the comparison groups.

Indicator	7th–9th Day		24 Weeks		48 Weeks	
	ALVR	non-ALVR	ALVR	non-ALVR	ALVR	non-ALVR
Ea/BSA, mmHg/mL	0.97 (0.89; 1.05)	1.01 (0.92; 1.09)	0.86 (0.78; 0.94) ##	0.98 (0.92; 1.05) *	0.93 (0.84; 1.02)	0.92 (0.86; 0.99) #
Ees/BSA, mmHg/mL	0.86 (0.77; 0.96)	1.13 (1.04; 1.23) **	0.70 (0.62; 0.78) ##	1.19 (1.10; 1.28) **	0.74 (0.65; 0.84) ##	1.20 (1.12; 1.28) **
Ea/Ees	1.27 (1.14; 1.39)	0.94 (0.85; 1.02) **	1.36 (1.23; 1.49)	0.84 (0.80; 0.88) **#	1.41 (1.25; 1.56) #	0.79 (0.74; 0.83) **##
SBPao, mmHg	98.9 (96.4; 101.5)	102.8 (100.2; 105.4) *	107.5 (104.3; 110.7) ##	109.7 (106.2; 113.1) ##	108.6 (105.7; 111.4) ##	112.4 (108.4; 116.3) ##
DBPao, mmHg	71.8 (69.6; 74.0)	72.7 (70.2; 75.2)	74.8 (72.0; 77.7) #	76.3 (73.8; 78.8) #	77.4 (75.3; 79.4) ##	77.1 (74.9; 79.3) #
cfPWV, m/s	7.8 (7.4; 8.3)	8.1 (7.6; 8.7)	7.7 (7.3; 8.2)	8.0 (7.5; 8.5)	7.6 (7.1; 8.0)	7.8 (7.3; 8.3)
QIMT, μm	798.2 (750.8; 845.6)	762.9 (722.9; 802.8)	758.2 (714.8; 801.6) ##	725.7 (692.0; 759.4) ##	735.7 (702.1; 769.2) ##	705.3 (669.7; 740.9) ##
β index	10.7 (9.5; 11.9)	9.3 (8.3; 10.2)	8.9 (8.2; 9.7) ##	8.2 (7.5; 8.9)	9.7 (8.5; 10.8) ##	8.5 (7.7; 9.2)
loc Psys, mmHg	101.8 (98.5; 105.1)	108.7 (105.8; 111.5) **	107.6 (105.3; 109.9) ##	113.1 (109.3; 116.8) *	111.2 (108.5; 113.8) ##	111.8 (108.7; 114.9)
loc Pdia, mmHg	68.7 (66.2; 71.2)	72.2 (70.2; 74.3) *	73.2 (71.1; 75.2) ##	75.3 (72.9; 77.7) #	75.9 (73.9; 77.8) ##	75.6 (73.8; 77.4) #

Note: the data are presented with 95% confidence interval; * *p* < 0.05, ** *p* < 0.01 is significance for intergroup differences; # *p* < 0.05, ## *p* < 0.01 is significance for intragroup differences on the baseline with subsequent visits; Ea/BSA is arterial elastance normalized to body surface area; Ees/BSA is end-systolic left ventricular elastance normalized to body surface area; Ea/Ees is left ventricular-arterial coupling index; SBPao is aortic systolic pressure; DBPao is aortic diastolic pressure; cfPWV is carotid-femoral pulse wave velocity; QIMT is quality intima-media thickness; loc Psys is local systolic blood pressure; loc Pdia is local diastolic blood pressure.

**Table 3 diagnostics-12-00770-t003:** Predictors of adverse left ventricular remodeling in patients after STEMI according to univariate analysis.

Indicator	β	Chi-Squared	*p*	RR (95% CI)
WC, cm	0.025	4.69	0.030	1.03 (1.002–1.05)
BNP, pg/mL	0.0009	6.50	0.011	1.001 (1.0002–1.002)
Abnormal BNP	0.88	10.11	0.001	2.41 (1.402–4.15)
EF, %	−0.057	14.57	0.0001	0.94 (0.92–0.97)
Ees/BSA, mmHg/mL/m^2^	−1.07	7.39	0.007	0.34 (0.16–0.74)
Ea/Ees	0.66	7.85	0.005	1.94 (1.22–3.08)
Abnormal Ea/Ees	0.82	10.42	0.001	2.27 (1.38–3.74)
loc Psys, mmHg	−0.020	4.07	0.044	0.98 (0.96–0.999)

Note: β is regression coefficient; *p* is significance; RR is relative risk; CI is confidence interval; WC is waist circumference; BNP is brain natriuretic peptide; EF is ejection fraction; Ees/BSA is end-systolic left ventricular elastance normalized to body surface area; Ea/Ees is Ea/Ees is left ventricular-arterial coupling index; loc Psys is local carotid systolic blood pressure.

**Table 4 diagnostics-12-00770-t004:** A multivariate model for the development of ALVR in STEMI patients.

Indicator	β	Chi-Squared	*p*	RR (95% CI)
WC, cm	0.024	4.11	0.042	1.02 (1.001–1.05)
Abnormal BNP	0.59	4.50	0.033	1.81 (1.05–3.13)
Abnormal Ea/Ees	0.68	5.45	0.020	1.96 (1.11–3.46)

Note: β—regression coefficient, *p*—significance, RR—relative risk, CI—confidence interval. WC, waist circumference; BNP, brain natriuretic peptide; Ea/Ees, left ventricular-arterial coupling index.

## Data Availability

The data presented in this study are available on request from the corresponding author. The data are not publicly available due to ethical reasons.

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
