# Peer review of "Prediction of Adverse Post-Infarction Left Ventricular Remodeling Using a Multivariate Regression Model"

_diagnostics, 2022, doi:10.3390/diagnostics12030770_

Round 1

Reviewer 1 Report

After a slow and critical reading of the present work, I would like to comment that, in my opinion, it is a very interesting study, with important clinical implications. The authors aimed to develop a model for predicting adverse remodeling in patients who had ST-segment elevation myocardial infarction. The article is well-structured, very interesting and the results are presented in an appropriate manner, being clear and transparent. The statistical analysis is also well done. I consider that the study is valuable and sound and can be published after some minor revisions. I would like to ask the authors if they have studied patients treated with ARNI and SGLT2 inhibitors?

Author Response

Thank you for your review. Patients have not been treated with ARNI and SGLT2 inhibitors.

Reviewer 2 Report

Valentin Oleynikov et al tried to predict adverse post-infarction left ventricular remodelling (ALVR) using a multivariate regression model in 125 STEMI patients. They defined ALVR as increase of end-diastolic volume index > 20% or increase of end-systolic volume index > 15%. Predictors for ALVR was high waist circumference, high BNP, increased BNP, low end-systolic left ventricular elastance normalized to body surface area, high left ventricular-arterial coupling index and low “local systemic” systolic blood pressure.

The underlying study reveals very interesting details: First, ALVR was very common (~50%), which is impressive in my eyes. Furthermore, patients with ALVR had significantly more long-term complications. Furthermore, the resulting predictors lead to several conclusions: abdominal fat (as expressed by waist circumference may be more important that BMI; BNP is a good long-term predictor; left ventricular elastance and left ventricular arterial coupling index may predict outcome; and carotid blood pressure may be most predictive of ALVR.

However, the current version of the manuscript is currently not adequately written and there are several points in the analysis that could be optimized.

First, the analysis lacks some important known risk factors. For example, I recommend to force left-ventricular ejection fraction into the multivariable analysis. Furthermore, in my opinion it is not statistically correct to include both the metric BNP value and condition of “increased BNP” into the multivariable analysis as these parameters are interdependent.

Second, but most importantly, the manuscript lacks severe organization and clarity. The abstract does not contain essential information, such as the definition of ALVR, demographical data, prevalence of ALVR and complications between both groups. Furthermore, all significant predictors in univariable analysis should be reported. In my opinion, there is no need to list all the examined parameters within the abstract. The sections “Materials and Methods” and “Results” are mixed up; demographics are presented even before the statistical methodology is explained. I recommend to move all the methodological parts, such as the paragraph below Figure 3 (“Univariate and multirvariate…”) into the results section. Furthermore, for example the third and forth paragraph of the Methods section should be moved to the Results section. Finally, I recommend to report the association between ALVR and long-term complications as one of the main results.

Minor comments:

  • Figure 1 is logical per definition of ALVR and may be moved to the supplemental appendix, except for EF.
  • Loc Psys should be defined as carotid systolic pressure (or am I wrong?)
  • The multivariate regression model should include either “X_1” or “X1” parameters (either leave or skip the “_”)
  • ALVR should be defined in the abstract and it should be defined only once in the text.
  • A study flowchart is missing.
  • The rate of primary lysis is pretty high. The authors should comment on this topic in the discussion.
  • The authors write that they would list all results with 95% confidence interval, without giving any CI in Table 1.
  • First paragraph of Results: I would not consider ALVR as “symptom”.
  • I recommend rewriting the first sentence of the manuscript (“CHF is an important topic in cardiology, […]”).

Author Response

First, the analysis lacks some important known risk factors. For example, I recommend to force left-ventricular ejection fraction into the multivariable analysis. 

Response: According to the univariate analysis, EF has proved to be a predictor of adverse left ventricular remodeling. The result was added to table 3. However, in a multivariate regression analysis, the combination of EF with other variables did not predict the development of adverse cardiac remodeling. The multivariate model presented in the article, including waist circumference, abnormal BNP, and Ea/Ees index, is the most informative among the studied parameters.

Furthermore, in my opinion it is not statistically correct to include both the metric BNP value and condition of “increased BNP” into the multivariable analysis as these parameters are interdependent.

Response: In this study, a bimodal distribution was used: at the BNP level <100 pg/ml, the value "0" was assigned; at the BNP level ≥100 pg/ml, the value "1" was assigned. When conducting a multivariate regression analysis, only one of the options for presenting BNP was simultaneously included in the model: absolute BNP values ​​as a continuous data series or BNP values ​​as a bimodal distribution. Table 3 contains both of these parameters, since it presents the indicators of one-way analysis. Univariate analysis describes the relationship between a function and one variable, that is, each of the presented indicators was tested separately.

Second, but most importantly, the manuscript lacks severe organization and clarity. The abstract does not contain essential information, such as the definition of ALVR, demographical data, prevalence of ALVR and complications between both groups. Furthermore, all significant predictors in univariable analysis should be reported. In my opinion, there is no need to list all the examined parameters within the abstract.

Response: We have included the required information in the abstract, removed the enumeration of all the studied parameters.

The sections “Materials and Methods” and “Results” are mixed up; demographics are presented even before the statistical methodology is explained. I recommend to move all the methodological parts, such as the paragraph below Figure 3 (“Univariate and multirvariate…”) into the results section.

Response: We moved all demographic data from the Materials and Methods to the Results. We have changed the wording of the first sentence.

Furthermore, for example the third and forth paragraph of the Methods section should be moved to the Results section.

Response: Corrected.

I recommend to report the association between ALVR and long-term complications as one of the main results.

Response: It is indicated in the manuscript.

Minor comments:

Figure 1 is logical per definition of ALVR and may be moved to the supplemental appendix, except for EF.

Response: Figure 1 has been corrected as recommended. A supplement has been added. Figure 1 was renamed to Figure 2, as another Figure 1 "Flow Diagram" was added.

Loc Psys should be defined as carotid systolic pressure (or am I wrong?)

Response: That’s correct.

The multivariate regression model should include either “X_1” or “X1” parameters (either leave or skip the “_”)

Response: Was corrected as «Х1».

ALVR should be defined in the abstract and it should be defined only once in the text.

Response: Corrected.

A study flowchart is missing.

Response: Flowchart of the study was added as figure 1.

The rate of primary lysis is pretty high. The authors should comment on this topic in the discussion.

Response: The term "lysis" is not clear to the authors in the context of the problem under discussion. The authors are ready to comment on this topic after clarification.

The authors write that they would list all results with 95% confidence interval, without giving any CI in Table 1.

Response: Corrections have been made to Table 1.

First paragraph of Results: I would not consider ALVR as “symptom”.

Response: The sentence was corrected.

I recommend rewriting the first sentence of the manuscript (“CHF is an important topic in cardiology, […]”).

Response: The sentence was corrected.

Reviewer 3 Report

The present study aimed to search for early predictors and develop a model for predicting adverse remodeling in patients who had ST-segment elevation myocardial infarction (STEMI)

The study is overall interesting, however, some issues must be addressed.

"The age of the patients included in the study was 51.2±8.8 years, men prevailed – 109 patients (87.2%). The body mass index was 27.4±3.7 kg/m2. Symptoms of abdominal obesity have been diagnosed in 74 patients (59.2%) [11]. Coronary heart disease has been noted in 21 patients in the history (16.8%). Cardiovascular diseases heredity has been burdened in 51 patients (40.8%) and 80 patients being smokers (64%). Arterial hypertension has been observed in 61.6% (n=77) with a disease duration of 6.4±5.5 years. The mean level of systolic blood pressure (SBP) was 115.5±10.9 mmHg and that of diastolic blood pressure (DBP) – 73.9±8.0 mmHg. Prior to the STEMI, antihypertensive therapy has been taken regularly by 16 patients (20.8%), and irregularly – by 24 patients (31.2%); 37 patients (48%) have been not treated.
Primary percutaneous coronary intervention has been performed in 56 patients (44.8%), pharmacoinvasive strategy – in 68 cases (54.4%). One patient has only received thrombolytic therapy at the pre-hospital stage." should be moved from the Material and Method section to Results.

Abbreviations used in Tables and Figures should be defined in their descriptions.

Limitations of the study should be highlighted.

Author Response

The specified fragment of the text was transferred from the Materials and Methods to the Results.

“The abbreviations used in tables and figures should be deciphered in their description,” - was added.

The limitations of the study were added.

Round 2

Reviewer 2 Report

In my opinion, the manuscript is still not suitable for publication.

  • The authors did not address some of my comments:
    • The authors still write "local systolic arterial pressure" although it is not clear where this pressure is measured.
    • The authors still use the same variable twice in the univariable analysis.
    • With "lysis therapy" I refer to "thrombolytic therapy". The authors should comment the fairly high rate of thrombolytic therapy.
  • In the abstract, results and methods are still mixed up (e.g. "A multifactorial model has been created" in the Results section) and it contains a lot of unnecessary information. It is currently very difficult to read. Major results (such as ALVR and major complications) are missing in the manuscript and they are not discussed there. I recommend to completely rewrite the abstract.

Author Response

Thank you for your review

The authors still write "local systolic arterial pressure" although it is not clear where this pressure is measured. 

Answer: the definition is corrected

The authors still use the same variable twice in the univariable analysis.

Answer: one variable was deleted

With "lysis therapy" I refer to "thrombolytic therapy". The authors should comment the fairly high rate of thrombolytic therapy.

Answer: the rate of thrombolytic therapy is added

In the abstract, results and methods are still mixed up (e.g. "A multifactorial model has been created" in the Results section) and it contains a lot of unnecessary information. It is currently very difficult to read. Major results (such as ALVR and major complications) are missing in the manuscript and they are not discussed there. I recommend to completely rewrite the abstract.

Answer: the abstract was rewritten